# Study on the Strength and Failure Characteristics of Silty Mudstone Using Different Unloading Paths

**DOI:** 10.3390/ma16145155

**Published:** 2023-07-21

**Authors:** Jijing Wang, Hualin Zhang, Shuangxing Qi, Hanbing Bian, Biao Long, Xinbo Duan

**Affiliations:** 1School of Traffic & Transportation Engineering, Changsha University of Science & Technology, Changsha 410114, China; 18002010031@stu.csust.edu.cn (S.Q.); lxy215512@163.com (B.L.); dyw12323@163.com (X.D.); 2IMT Nord Europe, University Lille, F-59000 Lille, France; hanbing.bian@univ-lille.fr; 3ULR 4515-LGCgE Laboratoire de Génie Civil et Géo-Environnement, JUNIA, University Artois, F-59000 Lille, France

**Keywords:** silty mudstone, unloading path, mechanical properties, shear strength, failure morphology

## Abstract

To investigate the strength and failure characteristics of silty mudstone using different stress paths, silt-like mudstone specimens were subjected to triaxial unloading tests. The results indicate the following. (1) When subjected to equivalent initial deviator stress levels and differing confining pressures, the peak stress, residual stress, and elastic modulus, exhibited during unloading, increased concordantly with greater initial confining pressure. Both the peak strain and residual strain increased with rising initial confining pressure. The increase in peak strain and residual strain initially decelerated, then noticeably increased, before ultimately decreasing again. Additionally, the unloading failure time and strain rate demonstrated a negative correlation as the confining pressure increased. (2) Under different initial deviatoric stress conditions, the peak stress, residual stress, and residual strain, under unloading confining pressure conditions, decreased as the initial deviatoric stress levels elevated. Conversely, the peak strain and elastic modulus initially increased, then decreased under increasing initial deviatoric stress conditions. The unloading failure time and strain rate were both observed to decrease as the initial deviatoric stress levels increased. (3) Utilizing the Mohr stress circle enabled the characterization of the shear strength variation in the specimens during the unloading process. The cohesion and internal friction angle remained relatively consistent across the different unloading stress paths appraised, with cohesion being greater in path I versus path II, whereas the internal friction angle exhibited an inverse relationship. (4) The specimen failed during unloading due to lateral expansion caused by unloading confining pressure and collapse failure. The failure fracture surfaces predominantly manifested shear failure morphologies.

## 1. Introduction

Silty mudstone is a type of soft rock which is widely distributed in the south of China. With the expansion of infrastructure in south-west China, this type of rock is frequently encountered in engineering practices. For instance, during highway construction, as the rock mass is continuously excavated, the initial in situ stresses are “unloaded”, which may lead to the redistribution of stresses. This triggers unloading deformation and potential damage, including micro or macro cracks within the rock mass [1,2,3,4]; these consequences are similar to the well-known Excavation Damaged/Distributed Zone (EDZ) found in tunnel engineering. Over the past decade, there have been numerous landslides triggered by the failure of soft rock in China, such as the Anlesi landslide and the Dalixi landslide, the landslide that occurred in the Three Gorges Reservoir area, and the landslide that occurred along the Jinchuan–Xiaojin highway in Sichuan Province; these are typical examples of unloading deformation caused by mudstone cut slope excavation [5,6,7]. To ensure the safety of human beings and infrastructures, it is of great importance to investigate the strengths and failures of silty mudstone under unloading conditions to provide fundamental strength parameters for construction practitioners.

Thus far, numerous researchers have directed their attention towards studying the strength, deformation, and expansion characteristics of rock specimens under unloading confining pressure conditions. These studies have primarily been conducted using experimental approaches, yielding valuable research findings. Through a comparison of conventional and unloading triaxial tests, Wang et al. [8] discovered that lower peak deviatoric stress and axial strain levels were exhibited during the unloading triaxial test. Conversely, other mechanical parameters, such as the internal friction angle and triaxial compressive strength, were found to exhibit higher levels than those obtained during the conventional triaxial test. Similarly, Zhang et al. [9] examined the impact of different stress paths on coal specimen mechanics, and they found that greater unloading rates heightened damage risk. Comparing the triaxial unloading test with the conventional triaxial compression test, a reduction in the coal specimen’s cohesive force and an increase in internal friction were noted. Moreover, Wang et al. [10] analyzed how the confining pressure and unloading rate impacted crack propagation; the results indicated that under greater confining pressure, crack formation post-failure, and the rate of crack expansion, increased with higher unloading rates. Takeda et al. [11] examined the semi-permeability evolution of Wakkanai mudstones during cyclic loading and unloading, emphasizing the importance of considering the applied stresses when estimating argillite semi-permeability. In response to stress evolution induced by activities such as excavation and mining, Zhang et al. [12] executed a multi-level axial stress triaxial unloading test that defined the alteration law of mechanical parameters (friction, cohesion, and dilatancy angle) using an unloading factor representing rock specimen damage mechanisms. Current research indicates that during a triaxial confining pressure unloading test, the failure mode, crack propagation, and strength characteristics largely hinge on factors such as initial confining pressure, unloading rate, and unloading path. Furthermore, different rock types exhibit unique mechanical behaviors under triaxial unloading confining pressure conditions. Shale displays significant brittleness, which escalates as the unloading rate and confining pressure increases. As the loading rate increases, sandstone shows an increasing stress brittleness reduction coefficient, indicating that its surface brittle failure decreases gradually. With the increasing unloading rate, the rock’s brittle failure initially strengthens and subsequently weakens; abnormal brittle failure is detected when the unloading rate is high [13,14].

In addition, significant progress has also been made in the investigation concerning the failure characteristics of rocks under cyclic loading in the early and subsequent unloading stages. Fan et al. [15] conducted comprehensive unloading tests on sandstone using the true triaxial system, which combined a damage control cyclic loading path with an unloading confining pressure path. The results suggested that the prior cyclic loading damage had an effect on strength and deformation characteristics, energy conversion, and failure modes. Huang et al. [16] demonstrated that the size and unloading rate of the initial confining pressure had a significant impact on the failure mode and strain energy conversion (accumulation, dissipation, and release). Zhao et al. [17], through the experiment, obtained mechanical variables (instantaneous elastic strain, viscoelastic strain, transient plastic strain, viscoplastic strain), indicating a positive relationship between the instantaneous deformation modulus and creep stress. These studies highlight how the multi-stage loading and unloading process during a cycle enhances a soft rock’s resistance to transient elastic and plastic deformation. Chen et al. [18] conducted a triaxial test on sandstone under confining pressure unloading conditions, measuring permeability and acoustic emission signals before peak stress was reached. The results demonstrate a strong correlation between the stress–strain curve, permeability–axial strain curve, and the acoustic emission activity mode under unloading conditions. Gupta et al. [19] examined shale deformation using various stress paths, highlighting the significant influence of stress level and bedding plane orientation on microcrack geometry. Additionally, Dai and Zhang et al. [20,21] showed that the strain increment is significantly influenced by the unloading path, whereas the unloading rate of the confining pressure is comparatively less affected.

Due to the relatively low strength, and the substantial sensitivity to water, silty mudstone has long been considered an inappropriate engineering material. Few investigations on its mechanical strength have been reported, particularly during “unloading” process. Most existing research has primarily focused on the mechanical properties and failure morphology of hard rocks using different loading and unloading paths, with fewer studies considering the deformation and comprehensive effects of different loading and unloading paths on soft rocks [22,23,24]. Therefore, in this paper, silt-like mudstone specimens, composed of gypsum, barite powder, and nanomaterials, were used to conduct triaxial tests using different unloading paths so that we could investigate the influence of different initial confining pressures and deviatoric stresses on the strength and failure characteristics of silty mudstone. The goal of this paper is to provide information on the safe operation and slope stability of silty mudstone areas, providing theoretical references for understanding rock performance under complex loading and unloading conditions.

## 2. Materials 

### 2.1. Specimens Preparation

The silty mudstone used in this paper was collected from the cutting slope of the Longlang Expressway, Hunan Province. The rock was rigorously characterized prior to testing. The major chemical components included SiO_2_, Al_2_O_3_, Fe_2_O_3_, K_2_O, MgO, TiO_2_, and Na_2_O. Notably, the collective mass fraction of SiO_2_ and Al_2_O_3_ constituted approximately 80% of all the chemical components. The granular composition consisted mainly of quartz, feldspar, clay minerals, and a small amount of carbonate minerals.

The inherent defects and low homogeneity of natural silty mudstone have a substantial impact on test results. To mitigate the impact of initial states on results, similar material specimens exhibiting excellent homogeneity were prepared. Silt-like specimens, composed of Nano-TiO_2_, Nano-Al_2_O_3_, NTi, Nano-bentonite, gypsum, and barite powder mixed with water, bore a close resemblance to silty mudstone in terms of its physical and mechanical characteristics, as reported in previous studies [25,26,27]. The silt-like mudstone specimens were tested using a controlled moisture content of 17.5%, which is within the range found in original rock (14.6–21.5%). The specimens were of a standard size (Φ50 × H100 mm). 

To validate the initial homogeneity of similar materials, a RSM-SY5(T) non-metallic ultrasonic tester was used on the processed silt-like mudstone specimens, in accordance with the ASTM D2845-standard [28], as displayed in Figure 1a. Two transducers were set on each specimen while Vaseline was applied for enhanced contact, and the sensor was connected to transducers. The travel time of the ultrasonic waves was measured, and the longitudinal wave velocity was calculated using the specimen length. Figure 1b illustrated the longitudinal wave velocity distribution for silt-like mudstone specimens. The wave velocity, primarily concentrated at 2.20–2.30 km/s, fell within the range found in original rock. Selecting specimens with similar wave velocities aided in minimizing variability, thus the specimens demonstrating wave velocities within the range of 2.20–2.30 km/s were selected for further testing (Figure 2). The comparison between the basic properties of the selected silt-like mudstone and the original rock is presented in Table 1.

### 2.2. Test Apparatus

The test employed the DZSZ-150 multi-field coupling rock triaxial testing machine, as illustrated in Figure 3. 

The machine utilized a full-digital servo controller as its control system, which incorporates high-precision load sensors and displacement sensors to automatically and accurately control and display data such as test force, confining pressure, and specimen deformation.

The machine was equipped with 1500 kN axial actuators, along with a 70 MPa confining pressure boosting system. Additionally, it featured a 70 MPa pore pressure boosting control system. The control panel allowed for the manual adjustment of the loading and unloading rate, pore pressure, back pressure, and axial pressure. The system was designed to collect test data automatically. The confining pressure in this system was generated by the oil pump.

## 3. Test Methods

This study employed non-standard loading and unloading pathways that mirrored actual excavation scenarios involving silty mudstone cutting slopes. Therefore, two unloading paths are considered in this experiment, as follows. (1) The first path involves the application of an initial deviatoric stress under unloading confining pressure conditions (i.e., the specimen is loaded until a particular level of predetermined deviatoric stress and confining pressure is reached). Subsequently, the deviatoric stress is kept constant while the confining pressure is gradually released at a specific rate until the specimen fails. This typically happens during the early stages of excavation when the horizontal confining pressure is reduced while the deviatoric stress caused by overburdening remains constant. (2) The second path concerns the application of a confining pressure that is left unchanged throughout the whole process. The specimen is then loaded to different levels of deviatoric stress and the confining pressure is unloaded until the specimen fails. This occurs in deep excavation scenarios, wherein the confining pressure remains steady due to the overlying rock layers, but the deviatoric stress varies due to changes in terms of overburdening or other external loading conditions [29,30].

The test procedure was conducted as follows: first, the confining pressure was applied in a stress-controlled manner at a rate of 0.075 MPa/s until it reached a predetermined value (20 MPa, 30 MPa, 40 MPa, and 50 MPa). Subsequently, with the confining pressure held steady, the axial pressure was applied at the same rate of 0.075 MPa/s to a predetermined axial pressure value (35 MPa, 45 MPa, 55 MPa, and 65 MPa). Finally, the confining pressure was incrementally unloaded at a rate of 0.075 MPa/s, while the axial pressure was kept constant until the specimen failed (Table 2).

## 4. Test Results and Discussions

### 4.1. Stress Path I

The stress–strain relationship curve of the silt-like mudstone when stress path I was used is depicted in Figure 4. It exhibits certain commonalities under different initial confining pressures when the deviatoric stress level was set to 15 MPa. More specifically, during this axial loading stage, the stress–strain curves demonstrate linear deformation, indicating the specimens’ linear elastic behavior. During the unloading stage, the axial strain continues to increase as the deviatoric stress increases, and a significant increase in elastic modulus is observed compared with the axial loading process. Post-failure, the stress continuously decreases, as the strain concurrently amplifies. And residual strain occurs at this stage, indicating the presence of apparent plastic flow features in the specimen.

The influence of the initial confining pressure on mechanical characteristics when unloading path I was used is depicted in Figure 5 and Figure 6, and the values are presented in the graph. As can be observed in the figure, for initial confining pressures of 20 MPa, 30 MPa, 40 MPa, and 50 MPa, the specimens display strains of 0.84%, 0.44%, 1.59%, and 0.87% until the unloading pressure was reached. The corresponding peak stresses and strains were 18.43 MPa (1.27%), 27.71 MPa (0.675%), 37.46 MPa (2.40%), and 40.41 MPa (0.96%), respectively. Likewise, the results for the elastic modulus are 6.74 GPa, 8.51 GPa, 10.01 GPa, and 14.98 GPa, respectively. 

The results reveal that the deviatoric stress at which failure occurs during unloading increases with higher levels of initial confining pressure. However, increases in deviatoric stress diminish as the confining pressure increases. The strain associated with unloading failure exhibits a trend of initially decreasing slowly and then increasing significantly as the initial confining pressure increases. The elastic modulus shows a notable increase with a higher initial unloading confining pressure, presenting a decelerating, then accelerating trend.

Post-unloading failure, the residual stresses and strains were 14.18 MPa (1.84%), 22.53 MPa (1.44%), 34.78 MPa (2.87%), and 37.87 MPa (2.22%), respectively. Notably, it demonstrated the marked increase in residual strain when the initial confining pressure was 40 MPa. Our interpretation focuses on the increased rock compression at higher confining pressures, which can lead to considerable strain prior to unloading failure. In particular, at this pressure, active micro-crack initiation and propagation are likely to contribute to increased plastic deformation, thus resulting in larger residual strain. These findings suggest a consistent rise in residual stress with increasing confining pressure. Although the specimens display an overall rising trend in residual strain, some fluctuations occur due to the compression density effect caused by the initial confining pressure. As the specimen becomes denser, the residual strain tends to increase. After the specimen fails, there is a certain randomness in the fracture surface and failure morphology, and this characteristic can notably affect the residual stress and strain.

The Mohr stress circle is plotted in accordance with the stress-strain curves of the silt-like mudstone under varying initial deviatoric stress and constant confining pressure conditions, as observed in Figure 7. The shear strength envelope of the silt-like mudstone specimen, as a function of the stress path I unloading confining pressure process, can be obtained as follows:(1)$$\tau =0.419\sigma +0.548,\;R^{2}=0.983$$
where *τ* is the shear stress and *σ* is the positive principal stress. Using Equation (1), the cohesive force *c* of the silt-like mudstone specimen is 0.548 MPa, and the internal friction angle *φ* is 26.151°.

From the left to right, Mohr stress circles represent a range from lower to higher initial confining pressure, exhibiting substantial differences. The process of unloading confining pressure shifted the Mohr’s circle towards the strength envelope. And Mohr stress circle expand as the confining pressure increases, indicating a decreased likelihood of failure in specimens.

For specimens under unloading path I, both the peak and residual stress increase with the initial confining pressure. Peak and residual strain decrease slowly as the initial confining pressure increases, then, they increase significantly and finally drop, both of which follow a similar pattern. The unloading failure time and strain rate are inversely proportional to the confining pressure increase. A higher initial unloading confining pressure corresponds to a substantial increase in elastic modulus.

### 4.2. Stress Path II

The stress–strain relationship curves for the silt-like mudstone, obtained using stress path II, are depicted in Figure 8. They exhibit consistent trends, demonstrating linear deformation characteristics when the initial confining pressure is maintained at 40 MPa and the deviatoric stress is increased to 10 MPa, 15 MPa, 20 MPa, and 25 MPa, respectively. This indicates linear elastic properties at this stage. As the confining pressure is unloaded, the axial strain continues to increase as the deviatoric stress increases. Upon failure of the silt-like mudstone specimen, the stress diminishes, causing the stress–strain curve to transition into the post-peak stage, which is characterized by a progressive decline in stress and a corresponding increase in strain. This stage, which exhibits significant plastic flow characteristics within the specimen, is where residual strain observed.

The variation in mechanical characteristics is due to initial deviatoric stress, as displayed in Figure 9 and Figure 10, and the mechanical parameters for unloading path II are illustrated in the graph. From the figure, it is evident that under a constant confining pressure of 40 MPa, with initial deviatoric stresses of 10 MPa, 15 MPa, 20 MPa, and 25 MPa, respectively, the peak stresses and strains are 47.07 MPa (1.620%), 37.46 MPa (2.182%), 32.81 MPa (1.740%), 25.80 MPa (1.736%). The results for the elastic modulus are 4.62 GPa, 9.94 GPa, 3.31 GPa, 3.12 GPa, respectively. 

The results indicate that peak stress decreases with an increase in initial deviatoric stress, whereas the peak strain exhibits a trend of initially increasing and subsequently decreasing. The elastic modulus demonstrates a substantial increase with higher initial deviatoric stress, followed by a considerable decrease, and finally a slow decrease. The residual stresses and strains were 42.27 MPa (3.228%), 34.78 MPa (3.024%), 28.70 MPa (2.153%), and 22.68 MPa (1.249%) after the failure of the specimen. Both the residual stress and strain exhibited a decreasing trend with increasing initial deviatoric stress.

Based on the stress–strain curve of silt-like mudstone under varying deviatoric stress conditions at a constant confining pressure, the Mohr stress circle can be plotted as shown in Figure 11. The shear strength envelope function of the silt-like mudstone specimen under stress path II can be obtained, as follows:(2)$$\tau =0.459\sigma +0.477,\;R^{2}=0.968$$
where *τ* is the shear stress and *σ* is the positive principal stress. Using Equation (2), the cohesive force c of the silt-like mudstone specimen is 0.477 MPa, and the internal friction angle *φ* is 26.335°.

Arranged from left to right, the Mohr stress circles indicate a progressively larger initial deviatoric stress in the specimens. The widening gap between the principal stresses within each Mohr stress circle suggests an escalating likelihood of failure as the deviatoric stress increases.

For the specimens subjected to the same confining pressure but varying deviatoric stresses when unloading path II was used, both the peak and residual stresses, as well as the residual strain, exhibited a decreasing trend with an increase in initial deviatoric stress; all displayed similar trends. The peak strain increased slowly with the increase in initial deviatoric stress, then, it significantly decreased. A negative correlation was observed between the unloading failure time and unloading strain rate in relation to the initial deviatoric stress. The elastic modulus demonstrated a notable trend of initially increasing and then decreasing with the increase in initial deviatoric stress under unloading confining pressure conditions.

### 4.3. Effect of Stress Path on Strength

During the triaxial unloading test, the specimen’s capacity to withstand the load is indeed influenced by the magnitude of the initial confining pressure and initial deviatoric stress. 

By increasing the deviatoric stress by reducing the confining pressure, the ultimate bearing capacity of the specimen is reduced to the ultimate stress level, leading to failure. 

Therefore, in this paper, two shear strength parameters, cohesion (*c*) and the internal friction angle (*φ*), have been selected to represent the shear strength fluctuations of specimens during unloading failure and to analyze the effect of the unloading path on shear strength [31,32,33]. The failure parameters of the specimens under different unloading paths are summarized in Table 3. The values of *c* and *φ* are comparably close for the specimens undergoing different unloading paths, and the cohesion of path I (0.548 MPa) exceeds that of path II (0.477 MPa). However, the internal friction angle of path I (26.151°) is slightly less than that of path II (26.335°).

## 5. Analysis of Macroscopic Characteristics of Specimens after Unloading Failure

### 5.1. Stress Path I

Figure 12 presents a typical image of silt-like mudstone specimens after unloading failure under path I. 

The image reveals that as the initial confining pressure increases, the failure morphology of the specimens becomes increasingly intricate, and they exhibit a distinct dilation feature, which is closely associated with the unloading confining pressure level of the specimens [9,34,35]. In the case of a low initial confining pressure, unloading failure mainly manifests as brittle shear failure, although, the structural integrity of the failed specimen remains intact. As the initial confining pressure increases, the failure morphology becomes more complex, and it is difficult to maintain the integrity of the failed specimen.

The findings can be summarized as follows: (1) under a low initial confining pressure (20 MPa), shear failure occurs with minimal tensile fractures, maintaining high specimen integrity with a few fine cracks on the fracture surface; (2) as the initial confining pressure increases (30 MPa), the specimen exhibits more tensile cracks in the axial stress direction and a noticeable degree of end fragmentation. Moreover, it demonstrates visible conjugate cracks and numerous microcracks, in which the primary fractured surface appears curved, which extends throughout the entire specimen. (3) as the initial confining pressure (40 MPa) further increases, the microcracks also continue to increase in number after the specimen undergoes shear failure. Penetrating cracks become less apparent, but a significant number of more pronounced microcracks emerge, which are widely distributed throughout the specimen; (4) lastly, after further increasing the initial confining pressure (50 MPa), the specimen under high confining pressures (40 MPa, 50 MPa) undergo significant fragmentation. The entire fractured surface attaches to a multitude of fine particles, resulting in the substantial scattering of particles due to the failure of specimens. The fractured surface exhibits a complex failure morphology, indicating the intact fragmentation of the specimen.

### 5.2. Stress Path II

Figure 13 illustrates a typical photograph of silt-like mudstone specimens after unloading failure when path II is used. 

It indicates that the unloading failure exhibits the pronounced failure characteristics of a fractured surface, which correlates with the confining pressure level of the specimens. Under low deviatoric stress conditions, it takes a relatively long time for the failure of specimen to occur under unloading confining pressure conditions. The failure primarily exhibits itself as brittle shear failure, and substantial integrity is maintained for the failed specimen. However, as the initial deviatoric stress increases, the integrity of the specimen after unloading failure deteriorates, and failure-induced cracks become more complex; in particular, severe end failure coupled with numerous localized micro-cracks may occur. Ultimately, extensive particle fragmentation occurs after failure when the initial deviatoric stress is 25 MPa.

The following can be concluded. (1) At an initial deviatoric stress of 10 MPa, the specimen mainly undergoes penetration fracture failure under constant deviatoric stress during unloading confining pressure, with noticeable tension fractures in the middle and upper sections, maintaining high integrity. (2) When the initial deviatoric stress increases to 15 MPa, the failure degree of the specimen end becomes distinctly visible, along with an observable penetration crack on its surface and the emergence of numerous end cracks. (3) As the initial deviatoric stress further increases to 20 MPa, post-shear failure microcracks proliferate throughout the specimen, with more pronounced local microcracks.

(4) When the initial deviatoric stress is 25 MPa, the specimen as a whole undergoes conspicuous shear failure, at which time the specimen completely fails. In particular, the load-bearing end demonstrates severe failure characteristics. This leads to the generation of a substantial number of broken particles, which is due to the strong frictional effect on the fractured surface of the entire specimen during the process of triaxial unloading failure.

After comparing the stress-strain relationship curves under different unloading conditions (path I and II) and the failure characteristics of the specimens, it is evident that the unloading failure is primarily dictated by the stress path, resulting in significant variations in terms of the macroscopic failure characteristics. In essence, the failure occurs due to the lateral expansion caused by the action of unloading confining pressure [36].

## 6. Conclusions

In this paper, an analysis was conducted to explore the strength and failure characteristics of silty mudstone that was subjected to various stress paths. The approach capitalized on silt-like mudstone specimens in triaxial unloading tests and it drew the principal conclusions, as follows:Under constant initial deviatoric stress and varying confining pressures, the silt-like mudstone exhibited continuous increases in peak stress, residual stress, and elastic modulus as the initial confining pressure increased during the unloading process. The peak strain and residual strain initially decreased gradually with the increase in initial confining pressure, subsequently displaying a notable increase before eventually decreasing. Additionally, the unloading damage time and unloading strain rate were found to be negatively correlated with the increase in confining pressure.Under different initial deviatoric stress conditions, the peak stress, residual stress, and residual strain under unloading confining pressure conditions exhibited a decreasing trend with an increase in initial deviatoric stress. The peak strain and elastic modulus initially demonstrated an increasing trend, followed by a subsequent decline, with an increase in initial deviatoric stress. Conversely, the unloading failure time and unloading strain rate decreased as the initial deviatoric stress increased.To characterize the shear strength variation of specimens, the cohesion and internal friction angle were obtained in accordance with the Mohr stress circle, the values of which were relatively close when both unloading stress path I and path II were used. Path I exhibited a larger cohesive force than path II, whereas the internal friction angle showed the opposite trend.The failure mechanism observed during the unloading of the specimens was characterized by a collapse-type failure, which may be primarily attributed to lateral expansion induced by the applied unloading confining pressure. The fractured surface predominantly exhibited shear failure. During the unloading process, when subjected to low initial confining pressure and low deviatoric stress conditions, the evolution of tension cracks was not prominently observed, and the specimens exhibited a high level of structural integrity. Conversely, under high initial confining pressure and high deviatoric stress conditions, the macroscopic development of cracks became more pronounced, leading to a greater degree of specimen fragmentation. Additionally, the end failure of the specimen became more evident.

## Figures and Tables

**Figure 1 materials-16-05155-f001:**
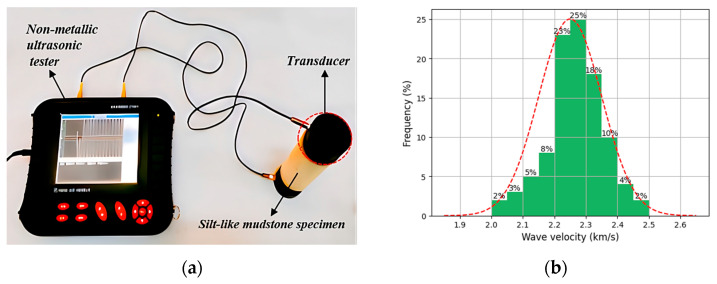
Ultrasonic longitudinal wave velocity measurements: (**a**) RSM-SY5(T) non-metallic ultrasonic tester; (**b**) wave velocity distribution for silt-like mudstone specimens.

**Figure 2 materials-16-05155-f002:**
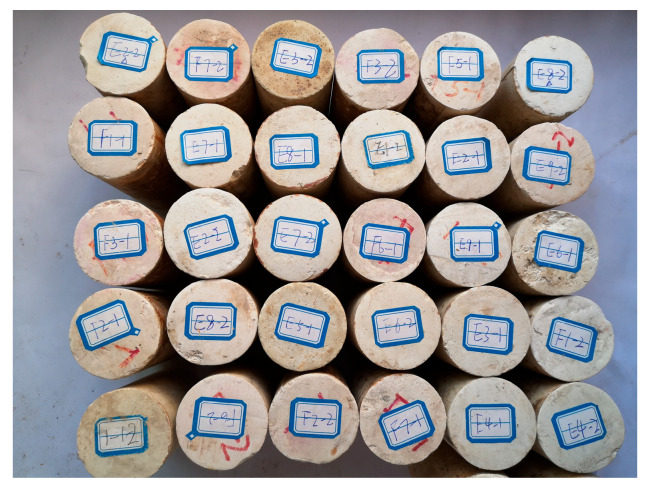
Selected silt-like mudstone specimens.

**Figure 3 materials-16-05155-f003:**
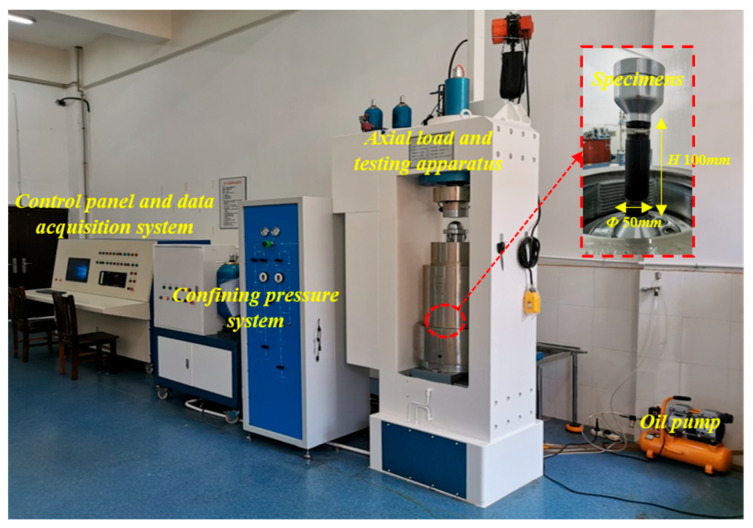
Test apparatus: DZSZ-150 multi-field coupling rock triaxial testing machine.

**Figure 4 materials-16-05155-f004:**
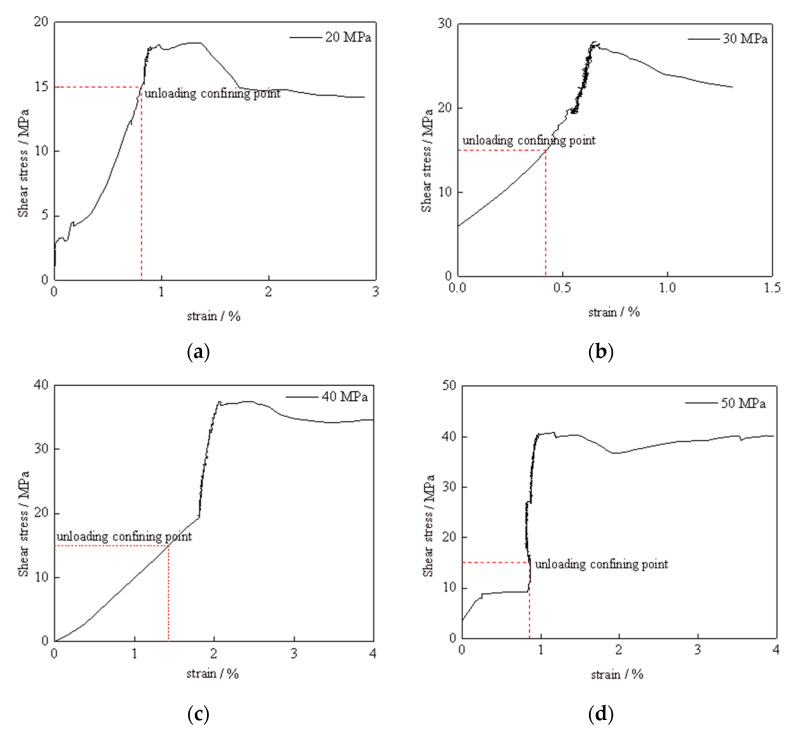
Stress–strain curve of silt-like mudstone specimens (path I): (**a**) initial confining pressure 20 MPa; (**b**) initial confining pressure 30 MPa; (**c**) initial confining pressure 40 MPa; (**d**) initial confining pressure 50 MPa.

**Figure 5 materials-16-05155-f005:**
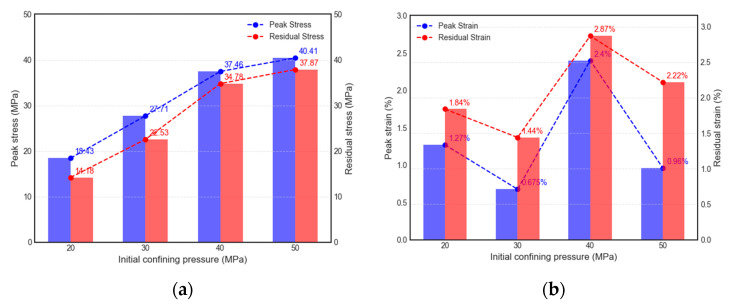
Influence of initial confining stress on mechanical parameters (path I): (**a**) the change law of peak stress and residual stress; (**b**) the change law of peak strain and residual strain.

**Figure 6 materials-16-05155-f006:**
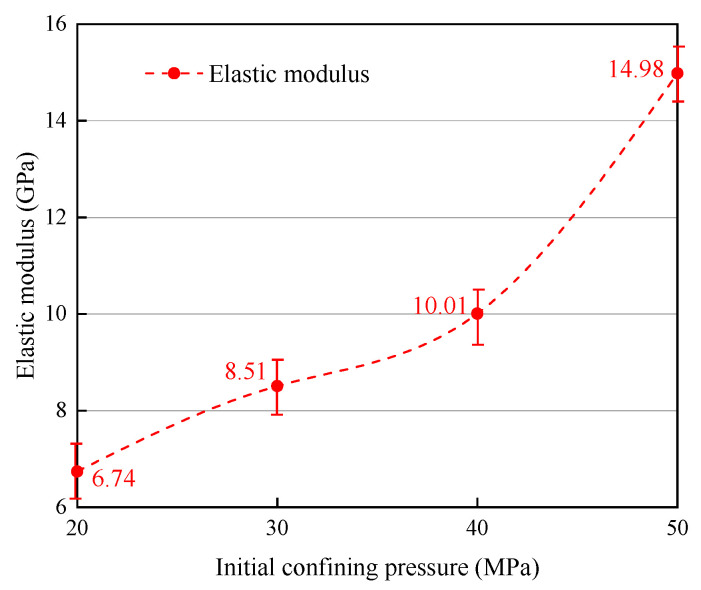
Influence of initial confining stress on elastic modulus (path I).

**Figure 7 materials-16-05155-f007:**
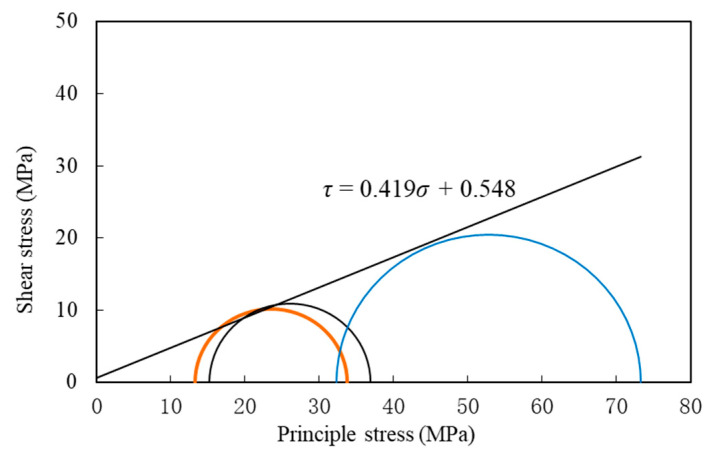
Mohr stress circle and strength envelope diagram of the silt-like mudstone specimen (path I).

**Figure 8 materials-16-05155-f008:**
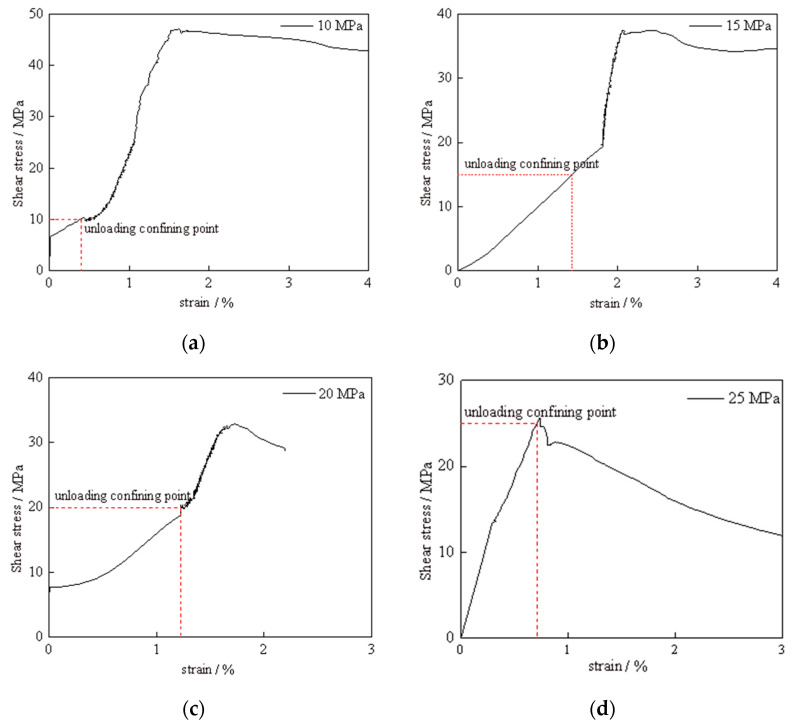
Stress–strain curve of silt-like mudstone specimen (path II): (**a**) deviatoric stress 10 MPa; (**b**) deviatoric stress 15 MPa; (**c**) deviatoric stress 20 MPa; (**d**) deviatoric stress 25 MPa.

**Figure 9 materials-16-05155-f009:**
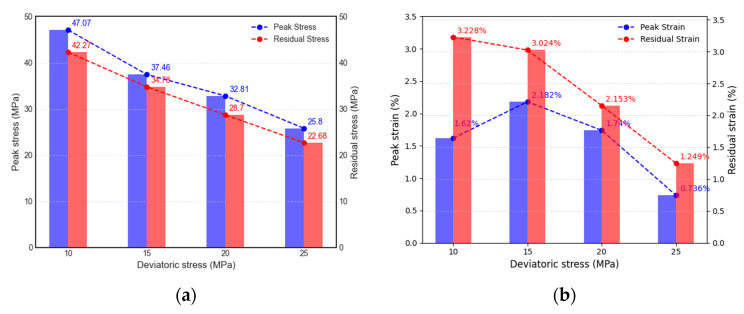
Influence of initial deviatoric stress on mechanical parameters (path Ⅱ): (**a**) the change law of peak stress and residual stress; (**b**) the change law of peak strain and residual strain.

**Figure 10 materials-16-05155-f010:**
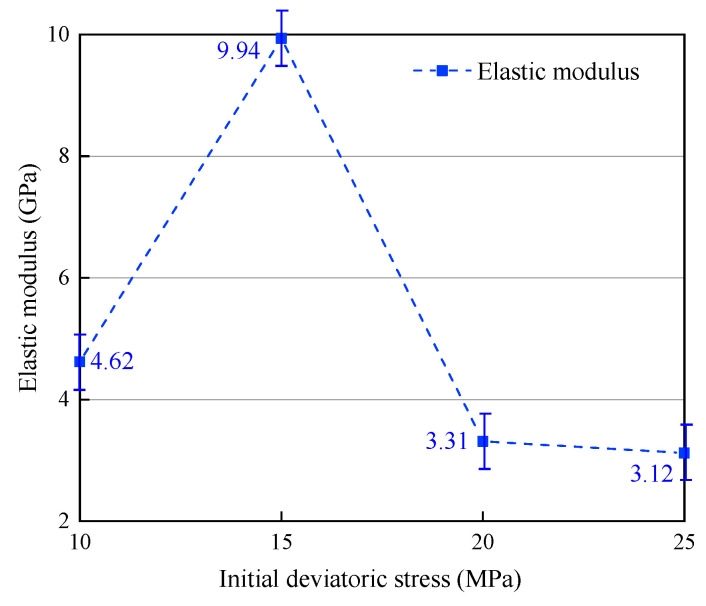
Influence of initial deviatoric stress on the elastic modulus (path Ⅱ).

**Figure 11 materials-16-05155-f011:**
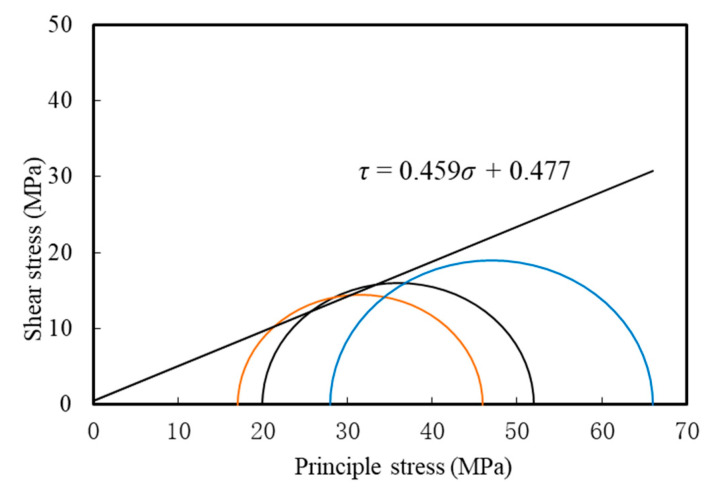
Mohr stress circle and strength envelope diagram of silt-like mudstone specimen (path II).

**Figure 12 materials-16-05155-f012:**
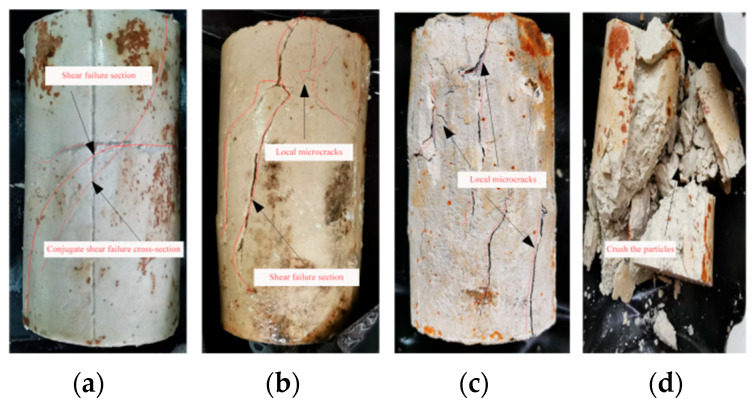
Typical failure morphology of silt-like mudstone specimens (Path I): (**a**) confining pressure 20 MPa; (**b**) confining pressure 30 MPa; (**c**) confining pressure 40 MPa; (**d**) confining pressure 50 MPa.

**Figure 13 materials-16-05155-f013:**
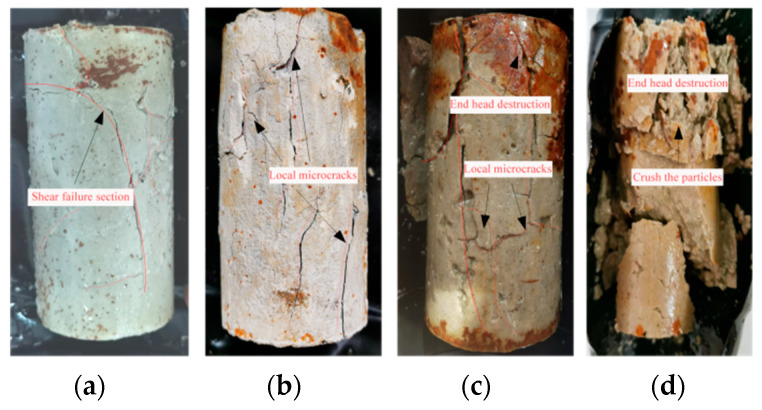
Typical failure morphology of silt-like mudstone specimens (Path II): (**a**) deviatoric stress 10 MPa; (**b**) deviatoric stress 15 MPa; (**c**) deviatoric stress 20 MPa; (**d**) deviatoric stress 25 MPa.

**Table 1 materials-16-05155-t001:** Comparison between basic properties of silt-like mudstone and original rock.

Parameters	Density (g/cm^3^)	Water Absorption Rate (%)	Swelling Rate (%)	Longitudinal Wave Speed (km/s)	Uniaxial Strength (MPa)	Tensile Strength (MPa)	Softening Coefficient
Original rock	2.15–2.34	2.14–8.16	0.14–0.18	1.75–2.85	6.9–23.8	0.3–1.4	0.4–0.66
Specimens	2.26	6.05	0.15	2.20–2.30	8.15	1.38	0.56

**Table 2 materials-16-05155-t002:** Test plan.

Path Number		Axial Pressure *σ*_1_ (MPa)	Confining Pressure *σ*_3_ (MPa)	Deviatoric Stress *σ*_1_–*σ*_3_ (MPa)	Axial Pressure after Damage (MPa)	Confining Pressure *σ*_3_ (MPa)
Path I	I-1	35	20	15	35	0.075 MPa/s unloading rate until damage
	I-2	45	30		45	
	I-3	55	40		55	
	I-4	65	50		65	
Path II	II-1	50	40	10	50	0.075 MPa/s unloading rate until damage
	II-2	55		15	55	
	II-3	60		20	60	
	II-4	65		25	65	

**Table 3 materials-16-05155-t003:** Failure strength parameters of silt-like mudstone specimens using different stress paths.

Stress Path	Cohesion *c* (MPa)	Internal Friction Angle *φ* (°)	Correlation Coefficient R^2^
Unloading path I	0.548	26.151	0.983
Unloading path II	0.477	26.335	0.968

## Data Availability

Not applicable.

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
