# Peer review of "Study on the Strength and Failure Characteristics of Silty Mudstone Using Different Unloading Paths"

_materials, 2023, doi:10.3390/ma16145155_

Round 1

Reviewer 1 Report

The article presents a study of the strength of silty siltstone under different stress paths.

The article deals with an interesting research topic, but in the reviewer's opinion, in its present form, it leaves a number of ambiguities in the methodological and material descriptions that need clarification and additions.

1 The language and style of the article are relatively correct, but due to the repetition of expressions in places and problems of interlineation, it is recommended to refine the text. Examples of errors:

-          table number 2 - interlineation errors cause problems in distinguishing stress paths.

-          L. 170 - 178: “The stress-strain curves of the silt-like mudstone exhibit certain commonalities, under different initial confining pressures and adding the deviatoric stress to 15 MPa, i.e., the stress-strain curves of the specimens demonstrate linear deformation, indicating their linear elastic properties in this axial loading stage; In the unloading stage, the axial strain of the specimens continues to increase as the increase of the deviatoric stress increases, and the elastic modulus shows a significant increase compared to the axial loading process; Upon the failure of the silt-like mudstone specimen, the stress gradually decreases, and the stress-strain curve enters the post-peak stage, where there is a tendency for the stress to continue decreasing while the strain keeps increasing.” This is one sentence that should be divided into several independent, shorter, coherent thoughts. As it stands, it is difficult to understand on first reading. There are more such long sentences in the article, it should be stylistically treated in this regard.

2. Materials characterization: Silty mudstone is a very general name for a rock that is not strictly characterized in the article. Its chemical and crumb composition is unknown. What were the parameters of the material at the time of the measurements, what was the moisture content of the tested samples - depending on the chemical and crumb composition of the rock, this can have a significant impact on the obtained results of the mechanical parameters.

3. Methods of testing:

In the reviewer's opinion, the non-standard sample loading and unloading paths adopted for the analysis should be explained. For what purpose were the adopted loading and unloading parameters and conditions used? What are the practical aspects of the applied test methodology?

L. 121-122: “To validate the initial homogeneity of similar materials, ultrasonic longitudinal wave velocity measurements were performed on the processed silty mudstone specimens, and the specimens with wave velocities of 2.20~2.30 km/s were screened for this test.” What was the methodology of the UPV testing? It would be interesting to see the results and their scatter for the tested mudstone.

The quality of the English language does not cause me any objections, but attention should be paid to the style of the language and the intricacy of some sentences, as they significantly affect the quality of the text. Examples of errors in this regard are indicated in the section for authors.

Author Response

Dear  Reviewer,

Thank you for giving us the opportunity to revise our manuscript. The comments are all valuable and very helpful for improving the quality of the manuscript. We have considered all comments carefully and have made relevant corrections in the revised manuscript. And the responses to all questions are in the attachment. 

Reviewer 2 Report

The work is interesting in a local sense, it can be seen from the cited sources, or is there no research in this field in other countries?

Line 118-120: This sentence is not clear. It is not clear what substances were tested “To overcome the influence of the initial state on results, similar materials, which have similar physical and mechanical properties to silty mudstone, are used in the testing process [25-27].”

What are the test materials made of? what is their chemical composition? where did these materials come from? are they ground? are they taken from somewhere?

Are they similar to the original rock? What is the composition of the original rock?

table 3 repeats the information already presented in the graphs.

Maybe you can explain the results obtained for residual strains when the initial confining pressure is 40MPa.

table 4 repeats the information already presented in the graphs.

In table 5 you give R2, is it a correlation coefficient or a coefficient of determination?

All research methods should be described in one chapter.

Author Response

(The authors gave the same response as above.)

Round 2

Reviewer 1 Report

Dear Authors

Since my comments were taken into account in the revised version of the article, in my opinion, it can be published in Materials in its current form.

Yours Sincerely,

Reviewer.